# High temperature singlet-based magnetism from Hund's rule correlations

Lin Miao[1,2], Rourav Basak[1], Sheng Ran[3], Yishuai Xu[1], Erica Kotta[1], Haowei He[1], Jonathan D. Denlinger[2], Yi-De Chuang[2], Y. Zhao [3,4], Z. Xu[3], J.W. Lynn[3], J.R. Jeffries[5], S.R. Saha[3,6], Ioannis Giannakis[7], Pegor Aynajian[7], Chang-Jong Kang [8], Yilin Wang[9], Gabriel Kotliar[8], Nicholas P. Butch[3,6] & L. Andrew Wray[1]

Uranium compounds can manifest a wide range of fascinating many-body phenomena, and are often thought to be poised at a crossover between localized and itinerant regimes for $5f$ electrons. The antiferromagnetic dipnictide $USb_2$ has been of recent interest due to the discovery of rich proximate phase diagrams and unusual quantum coherence phenomena. Here, linear-dichroic X-ray absorption and elastic neutron scattering are used to characterize electronic symmetries on uranium in $USb_2$ and isostructural $UBi_2$. Of these two materials, only $USb_2$ is found to enable strong Hund's rule alignment of local magnetic degrees of freedom, and to undergo distinctive changes in local atomic multiplet symmetry across the magnetic phase transition. Theoretical analysis reveals that these and other anomalous properties of the material may be understood by attributing it as the first known high temperature realization of a singlet ground state magnet, in which magnetism occurs through a process that resembles exciton condensation.

[1] Department of Physics, New York University, New York, NY 10003, USA. [2] Advanced Light Source, Lawrence Berkeley National Laboratory, Berkeley, CA 94720, USA. [3] NIST Center for Neutron Research, National Institute of Standards and Technology, Gaithersburg, MD 20899, USA. [4] Department of Materials Science and Engineering, University of Maryland, College Park, MD 20742, USA. [5] Materials Science Division, Lawrence Livermore National Laboratory, Livermore, CA 94550, USA. [6] Center for Nanophysics and Advanced Materials, Department of Physics, University of Maryland, College Park, MD 20742, USA. [7] Department of Physics, Applied Physics and Astronomy, Binghamton University, Binghamton, NY 13902, USA. [8] Department of Physics and Astronomy, Rutgers University, Piscataway, NJ 08854-8019, USA. [9] Department of Condensed Matter Physics and Materials Science, Brookhaven National Laboratory, Upton, NY 11973, USA. Correspondence and requests for materials should be addressed to L.A.W. (email: lawray@nyu.edu)

Uranium compounds can feature a fascinating interplay of strongly correlated and itinerant electronic physics, setting the stage for emergent phenomena such as quantum criticality, heavy fermion superconductivity, and elusive hidden order states[1–13]. The isostructural uranium dipnictides $UX_2$ (X = As, Sb, Bi) present a compositional series in which high near-neighbor uranium-uranium coordination supports robust planar antiferromagnetism ($T_N$~200K, see Fig. 1a, b)[7,8]. Of these, the $USb_2$ variant has received close attention due to the discovery of several unexplained low temperature quantum coherence phenomena at $T < 100K$[7,9–11], and a remarkably rich phase diagram incorporating quantum critical and tricritical points as a function of pressure and magnetic field[12,13]. However, the effective valence state of uranium and the resulting crystal field state basis defining the f-electron component of local moment and Kondo physics have not been identified.

Here, X-ray absorption (XAS) at the uranium O-edge and numerical modeling are used to evaluate the low energy atomic multiplet physics of $USb_2$ and $UBi_2$, revealing only $USb_2$ to have significant Hund's rule correlations. These investigations yield the prediction that $USb_2$ must be a uniquely robust realization of a singlet-ground-state magnet, in which magnetic moments appear via the occupation of low-energy excited states on a non-magnetic background (Fig. 1c). The evolution of crystal field symmetries and magnetic ordered moment across the antiferromagnetic phase transition is measured with linear dichroism (XLD) and elastic neutron scattering, confirming that the magnetic transition in $USb_2$ occurs through an exotic process that resembles exciton condensation.

## Results

**Electron configuration of uranium in $UBi_2$ and $USb_2$.** Unlike the case with stronger ligands such as oxygen and chlorine, there is no unambiguously favored effective valence picture for uranium pnictides. Density functional theory suggests that the charge and spin density on uranium are significantly modified by itinerancy effects[14,15] (see also Supplementary Note 1), as we will discuss in the analysis below, making it difficult to address this question from secondary characteristics such as the local or ordered moment. However, analyses in 2014–2016 have shown that resonant fine structure at the O-edge ($5d{\rightarrow}5f$ transition) provides a distinctive fingerprint for identifying the nominal valence state and electronic multiplet symmetry on uranium[16–19]. X-ray absorption spectra (XAS) of $UBi_2$ and $USb_2$ were measured by the total electron yield (TEY) method, revealing curves that are superficially similar but quantitatively quite different (Fig. 2a). Both curves have prominent resonance features at $h\nu$~100 and ~113 eV that are easily recognized as the 'R1' and 'R2' resonances split by the G-series Slater integrals[16]. Within models, these resonances are narrowest and most distinct for $5f^0$ systems, and merge as $5f$ electron number increases, becoming difficult to distinguish beyond $5f^2$ (see Fig. 2a (bottom) simulations). The $USb_2$ sample shows absorption features that closely match the absorption curve of $URu_2Si_2$[16], and are associated with the $J = 4$ ground states of a $5f^2$ multiplet. This correspondence can be drawn with little ambiguity by noting a one-to-one feature correspondence with the fine structure present in a second derivative analysis (SDI, see Fig. 2b).

The R1 and R2 resonances of $UBi_2$ are more broadly separated than in $USb_2$, and the lower energy R1 feature of $UBi_2$ is missing the prominent leading edge peak at $h\nu$~98.2 eV (peak-B), which is a characteristic feature of $5f^2$ uranium[16,17]. The $UBi_2$ spectrum shows relatively little intensity between R1 and R2, and the higher energy R2 resonance has a much sharper intensity onset. All of these features are closely consistent with expectations for a $5f^1$ multiplet, and the SDI curve in Fig. 2b reveals that the R1 fine structure of $UBi_2$ is a one-to-one match for the $5f^1$ multiplet. We note that a close analysis is not performed for R2 as it is influenced by strong Fano interference (see Supplementary

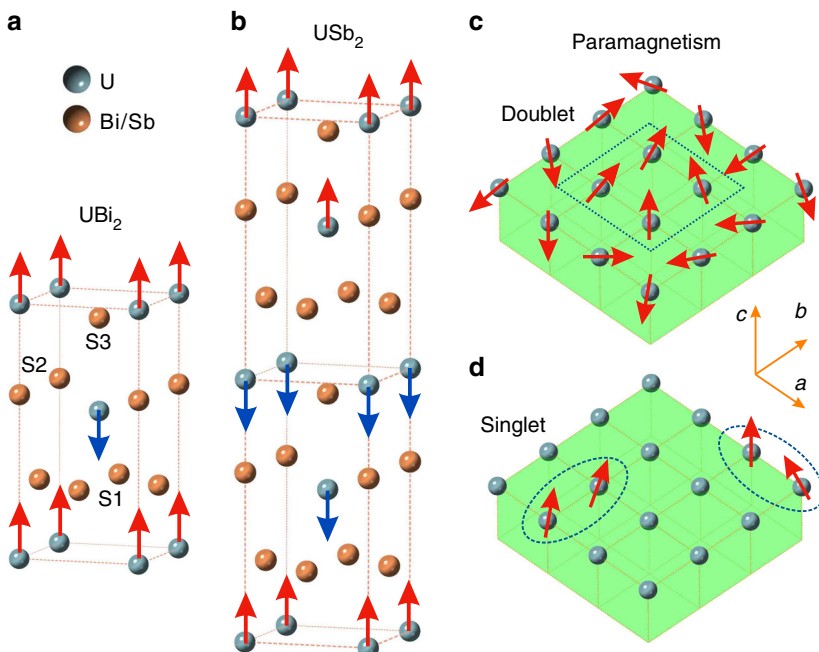

**Fig. 1** Singlet ground state magnetism and the ligand cage of $U(Bi/Sb)_2$. **a, b** The $U(Sb/Bi)_2$ crystal structure is shown with spins indicating the antiferromagnetic structure in $UBi_2$ ($T_N$~180 K) and $USb_2$ ($T_N$~203 K). The uranium atoms have 9-fold ligand coordination with base (S1), middle (S2), and pinnacle (S3) ligand layers as labeled in **a** with respect to the central uranium atom. **c, d** In-plane ferromagnetic nucleation regions are circled in **c** doublet and **d** singlet ground state magnetic systems. The singlet crystal field ground state has no local moment, causing much of the lattice to have little or no magnetic polarization

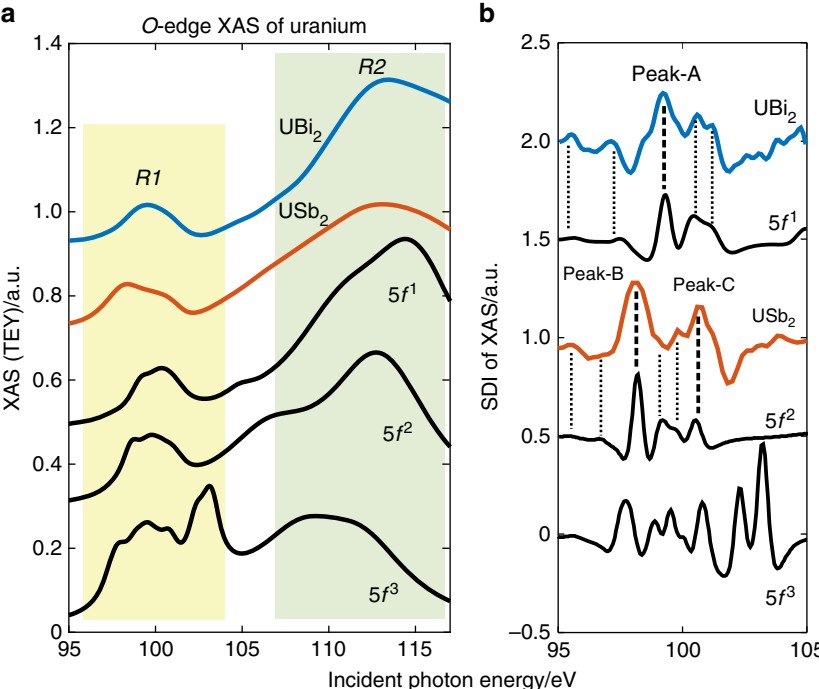

**Fig. 2** XAS fine structure and valence of $UBi_2$ and $USb_2$. **a** The x-ray absorption of $UBi_2$ and $USb_2$ on the O-edge of uranium is compared with (bottom) multiplets simulations for $5f^1$ ($U^{5+}$), $5f^2$ ($U^{4+}$), and $5f^3$ ($U^{3+}$). **b** A negative second derivative (SDI) of the XAS data and simulated curves, with drop-lines showing feature correspondence. Noise in the SDI has an amplitude comparable to the plotted line thickness, and all features identified with drop-lines were consistently reproducible when moving the beam spot. Prominent absorption features are labeled peak-A ($UBi_2$, $h\nu = 99.2$ eV), peak-B ($USb_2$, $h\nu = 98.2$ eV), and peak-C ($USb_2$, $h\nu = 100.8$ eV). Source data are provided as a Source Data file

| **Table 1 The CEF energy hierarchy in $USb_2$** | | | | |
|---|---|---|---|---|
| | **CEF(1) (20/33/33)** | **CEF(2) (33/33/33)** | **CEF(3) (50/33/33)** | **CEF(4) (80/130/130)** |
| $\Gamma_1$ (1) | 0 | 0 | 0 | 0 |
| $\Gamma_5$ (2) | 10.0 | 11.4 | 12.6 | 38.5 |
| $\Gamma_2$ (1) | 13.1 | 10.8 | 8.8 | 51.6 |
| $\Gamma_3$ (1) | 13.6 | 13.9 | 15.2 | 54.3 |
| $\Delta$CEF | 27.2 | 30.8 | 37.5 | 106.0 |

The energies in millielectron volts of low-lying $5f^2$ multiplet symmetries are shown for four crystal field parameter sets. Parameters in the first column (CEF(1)) follow the relative energy ordering suggested in ref. [8]. (S1 < S2-S3, as the S1 bond is relatively short), and are used for all simulations. The state symmetries are summarized in Supplementary Note 5, which includes an energy level diagram. $\Delta$CEF is defined as the gap between the highest energy $J = 4$ CEF state and the ground state. CEF parameters listed as (S1/S2/S3) for the sites defined in Fig. 1a. These values have units of millielectron volts, and define delta function potentials for Sb atoms in the (S1) base, (S2) middle, and (S3) c-axis pinnacle of the $Sb_9$ cage around each uranium atom. Specifically, the energy parameters indicate the energy added by a single Sb atom to an $m_j = 0$ f-orbital oriented along the U-Sb axis. Source data are provided as a Source Data file

Note 2). The lack of prominent $5f^2$ multiplet features suggests that the $5f^1$ multiplet state is quite pure, and the measurement penetration depth of several nanometers (see Methods) makes it unlikely that this distinction between UHV-cleaved $UBi_2$ and $USb_2$ originates from surface effects. However, the picture for $UBi_2$ is complicated by a very rough cleaved surface, which our STM measurements (see Supplementary Note 3) find to incorporate at least two non-parallel cleavage planes. Surface oxidation in similar compounds is generally associated with the formation of $UO_2$ ($5f^2$) and does not directly explain the observation of a $5f^1$ state.

We note that even with a clean attribution of multiplet symmetries, it is not at all clear how different the f-orbital occupancy will be for these materials, or what magnetic moment should be expected when the single-site multiplet picture is modified by band-structure-like itinerancy[10,11] (see also Supplementary Note 1). The effective multiplet states identified by shallow-core-level spectroscopy represent the coherent multiplet (or angular moment) state on the scattering site and its

surrounding ligands, but are relatively insensitive to the degree of charge transfer from the ligands[20].

Nonetheless, the $5f^1$ and $5f^2$ nominal valence scenarios have very different physical implications. A $5f^1$ nominal valence state does not incorporate multi-electron Hund's rule physics[21,22] (same-site multi-electron spin alignment), and must be magnetically polarizable with non-zero pseudospin in the paramagnetic state due to Kramer's degeneracy (pseudospin ½ for the $UBi_2$ crystal structure). By contrast in the $5f^2$ case one expects to have a Hund's metal with strong alignment of the 2-electron moment (see dynamical mean field theory (DFT + DMFT) simulation below), and the relatively low symmetry of the 9-fold ligand coordination around uranium strongly favors a non-magnetic singlet crystal electric field (CEF) ground state with $\Gamma_1$ symmetry, gapped from other CEF states by roughly 1/3 the total spread of state energies in the CEF basis (see Table 1). The $\Gamma_1$ state contains equal components of diametrically opposed large-moment $|m_J = +4>$ and $|m_J = -4>$ states, and is poised with no net moment by the combination of spin-orbit and CEF

interactions. This unusual scenario in which magnetic phenomena emerge in spite of a non-magnetic singlet ground state has been considered in the context of mean-field models[23–26], and appears to be realized at quite low temperatures (typically $T < \sim 10$ K) in a handful of rare earth compounds. The resulting magnetic phases are achieved by partially occupying low-lying magnetic excited states, and have been characterized as spin exciton condensates[23].

**Multiplet symmetry from XLD versus temperature.** To address the role of low-lying spin excitations, it is useful to investigate the interplay between magnetism and the occupied multiplet symmetries by measuring the polarization-resolved XAS spectrum as a function of temperature beneath the magnetic transition. Measurements were performed with linear polarization set to horizontal (LH, near $z$-axis) and vertical (LV, $a$–$b$ plane) configurations. In the case of UBi$_2$, the XAS spectrum shows little change as a function of temperature from 15 to 210K (Fig. 3a, b), and temperature dependence in the dichroic difference (XLD, Fig. 3b) between these linear polarizations is inconclusive, being dominated by noise from the data normalization process (see Methods and Supplementary Note 4). This lack of temperature dependent XLD is consistent with conventional magnetism from a doublet ground state. The XLD matrix elements do not distinguish between the up- and down-moment states of a Kramers

doublet, and so strong XLD is only expected if the magnetic phase incorporates higher energy multiplet symmetries associated with excitations in the paramagnetic state.

By contrast, the temperature dependence of USb$_2$ shows a large monotonic progression (Fig. 3c, d), suggesting that the atomic symmetry changes significantly in the magnetic phase. The primary absorption peak ($h\nu \sim 98.2$ eV, peak-B) is more pronounced under the LH-polarization at low-temperature, and gradually flattens as temperature increases. The LV polarized spectrum shows the opposite trend, with a sharper peak-B feature visible at high temperature, and a less leading edge intensity at low temperature. This contrasting trend is visible in the temperature dependent XLD in Fig. 3d, as is a monotonic progression with the opposite sign at peak-C ($h\nu \sim 100.8$ eV).

Augmenting the atomic multiplet model for $5f^2$ uranium with mean-field magnetic exchange (AM + MF) aligned to match the $T_N \sim 203$K phase transition (see Methods) results in the temperature dependent XAS trends shown in Fig. 3e. The temperature dependent changes in peak-B and peak-C in each linear dichroic curve match the sign of the trends seen in the experimental data, but occur with roughly twice the amplitude, as can be seen in Fig. 3d, f. No attempt is made to precisely match the $T > 200$K linear dichroism, as this is influenced by itinerant and Fano physics not considered in the model. The theoretical amplitude could easily be reduced by adding greater broadening on the energy loss axis or by fine tuning of the model (which has been

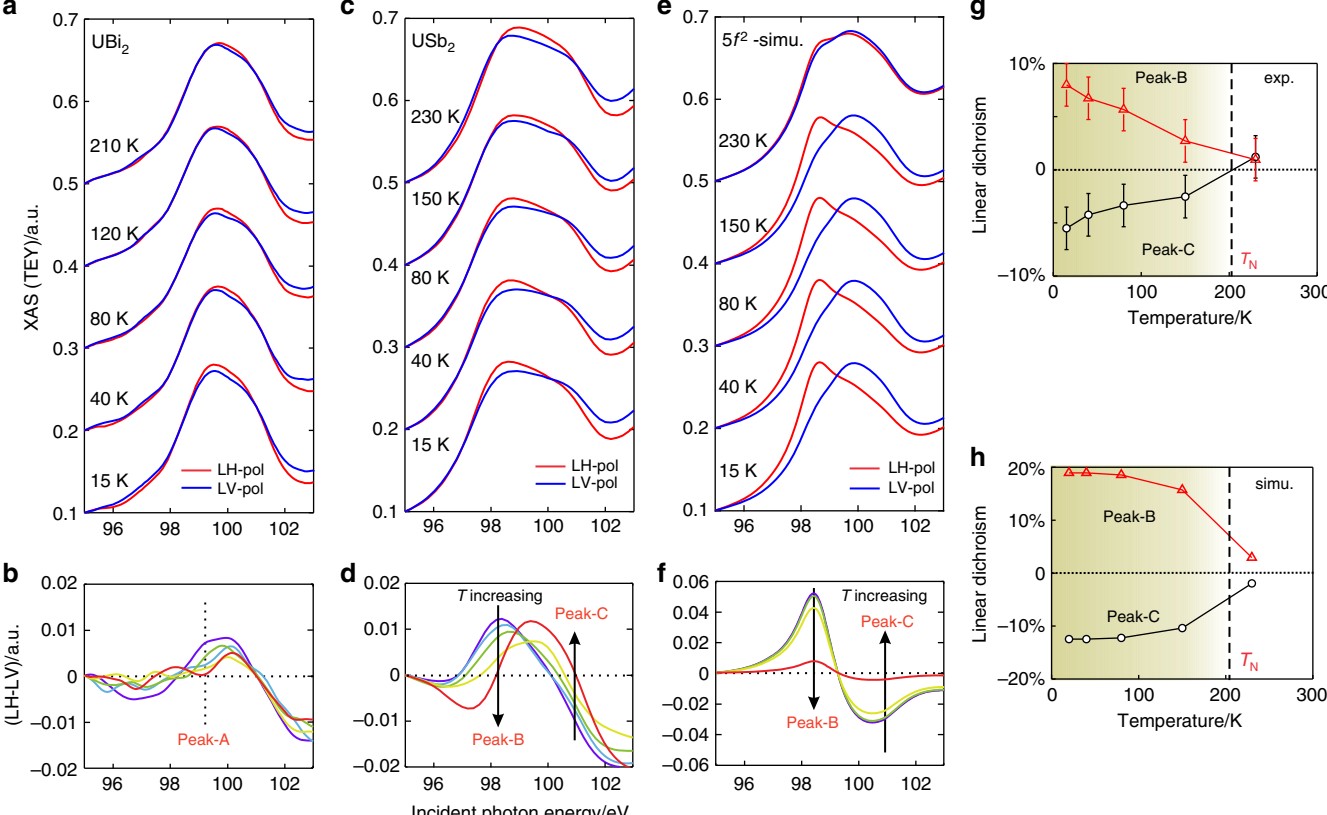

**Fig. 3** Temperature dependence of occupied f-electron symmetries. **a** The R1 XAS spectrum of UBi$_2$ is shown for linear horizontal (LH) and vertical (LV) polarizations. **b** The dichroic difference (LH-LV) is shown with temperature distinguished by a rainbow color order (15K (purple), 40K (blue), 80K (green), 120K (yellow), and 210K (red)). **c**, **d** Analogous spectra are shown for USb$_2$. Arrows in **d** show the monotonic trend direction on the peak-B and peak-C resonances as temperature increases. **e**, **f** Simulations for $5f^2$ with mean-field magnetic interactions. **g** A summary of the linear dichroic difference on the primary XAS resonances of USb$_2$, as a percentage of total XAS intensity at the indicated resonance energy ($h\nu = 98.2$ eV for peak-B, and $h\nu = 100.8$ eV for peak-C). Error bars represent a rough upper bound on the error introduced by curve normalization. **h** The linear dichroic difference trends from the mean field model. Source data are provided as a Source Data file. Shading in **g**, **h** indicates the onset of a magnetic ordered moment

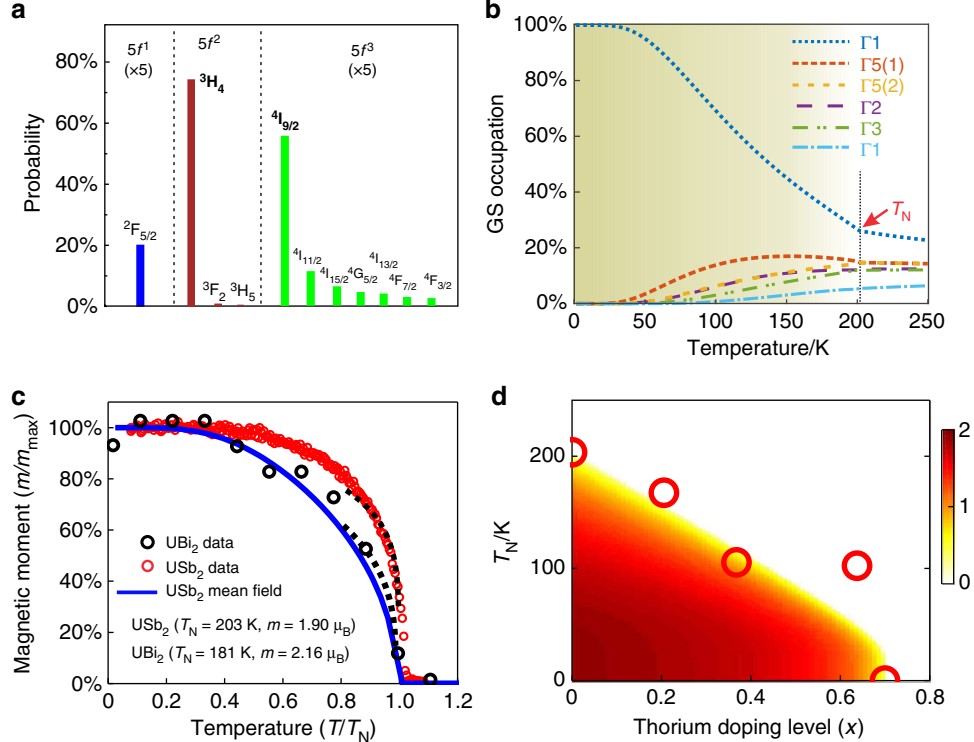

**Fig. 4** Electronic symmetry convergence in $USb_2$. **a** The partial multiplet state occupancy on uranium in $USb_2$ from DFT + DMFT numerics, with Hund-aligned symmetries highlighted in bold ($^3H_4$ and $^4I_{9/2}$). **b** Temperature dependence of the partial occupancy of different multiplet states within a $5f^2$ mean field model. In spite of a magnetic transition above 200K, roughly 1/3$^{rd}$ of the ground state convergence occurs in the range from 30–100K. The labeled CEF symmetries are only fully accurate in the high temperature paramagnetic state. Beneath the Néel temperature, the $\Gamma_1$ ground state is magnetically polarized by admixture with $\Gamma_2$. Shading indicates the onset of a magnetic ordered moment. **c** The ordered magnetic moment of (red circles) $USb_2$ and (black circles) $UBi_2$ from elastic neutron scattering. The mean field multiplet model for $USb_2$ is shown as a solid blue curve, and critical exponent trends near the phase transition are traced with dashed black lines representing $m(T) = m_{max}(1 - T/T_N)^\beta$. The $USb_2$ data are overlaid with a steep critical exponent trend of $\beta = 0.19$ indicating strong fluctuations, and the $UBi_2$ data are overlaid with the conventional 3D Ising critical exponent ($\beta = 0.327$). **d** The Néel temperature as a function of doping level in $U_{1-x}Th_xSb_2$ (red circles), and the simulated ordered moment in Bohr magnetons (renormalized to 62% as described in Methods; red-hot shading). Source data for all curves are provided as a Source Data file

avoided – see Methods). However, it is difficult to compensate for a factor of two, and the discrepancy is likely to represent a fundamental limitation of the non-itinerant mean field atomic multiplet model. Indeed, when the competition between local moment physics and electronic itinerancy is evaluated for $USb_2$ with dynamical mean field theory (DFT + DMFT), we find that the uranium site shows a non-negligible ~25% admixture of $5f^1$ and $5f^3$ configurations (Fig. 4a).

**Magnetic ordered moment and the nature of fluctuations.**
Compared with conventional magnetism, the singlet ground state provides a far richer environment for low temperature physics within the magnetic phase. In a conventional magnetic system, the energy gap between the ground state and next excited state grows monotonically as temperature is decreased beneath the transition, giving an increasingly inert many-body environment. However, in the case of singlet ground state magnetism, the ground state is difficult to magnetically polarize, causing the energy gap between the ground state and easily polarized excited states to shrink as temperature is lowered and the magnetic order parameter becomes stronger. Consequently, within the AM + MF model, many states keep significant partial occupancy down to $T < 100$K, and the first excited state (derived from the $\Gamma_5$ doublet) actually grows in partial occupancy beneath the phase transition (see Fig. 4b). Of the low energy CEF symmetries (tracked in

Fig. 4b), $\Gamma_5$ and $\Gamma_2$ are of particular importance, as $\Gamma_5$ is a magnetically polarizable Ising doublet, and $\Gamma_2$ is a singlet state that can partner coherently with the $\Gamma_1$ ground state to yield a z-axis magnetic moment (see Supplementary Note 5). These non-ground-state crystal field symmetries retain a roughly 1/3$^{rd}$ of the total occupancy at $T = 100$K, suggesting that a heat capacity peak similar to a Schottky anomaly should appear at low temperature, as has been observed at $T < ~50$K in experiments (see the supplementary material of ref. [10]). Alternatively, when intersite exchange effects are factored in, the shrinking energy gap between the $\Gamma_1$ and $\Gamma_5$ CEF states at low temperature will enable Kondo-like resonance physics and coherent exchange effects that are forbidden in conventional magnets.

Critical behavior at the Néel transition should also differ, as the phase transition in a singlet-ground-state magnet is only possible on a background of strong fluctuations. Measuring the ordered moment as a function of temperature with elastic neutron scattering (Fig. 4c) reveals that the $UBi_2$ moment follows a trend that appears consistent with the $\beta = 0.327$ critical exponent for a 3D Ising system[27]. The order parameter in $USb_2$ has a sharper onset that cannot be fitted sufficiently close to the transition point due to disorder, but can be overlaid with an exponent of $\beta \sim 0.19$, and may resemble high-fluctuation scenarios such as tricriticality ($\beta = 0.25$[28,29]). This sharp onset cannot be explained from the AM + MF model (blue curve in Fig. 4c), as mean field models that replace fluctuations with a static field give large critical

exponents such as $\beta = 0.5$ and unphysically high transition temperatures in systems where fluctuations are important. Another approach to evaluate the importance of fluctuations is to lower the Néel temperature by alloying with non-magnetic thorium (Th), as $U_{1-x}Th_xSb_2$ (see Fig. 4d), thus quenching thermal fluctuations at the phase transition. Performing such a growth series reveals that the magnetic transition can be suppressed to $T_N \sim 100$ K, but is then abruptly lost at $x \sim 0.7$, consistent with the need for fluctuations across a CEF gap of $k_B T_N \sim 10$ meV, which matches expectations from theory for the energy separation between $\Gamma_1$ and $\Gamma_5$ (see Table 1 and Methods).

## Discussion

In summary, we have shown that the $USb_2$ and $UBi_2$ O-edge XAS spectra represent different nominal valence symmetries, with $USb_2$ manifesting $5f^2$ moments that are expected to create a Hund's metal physical scenario, and $UBi_2$ showing strong $5f^1$–like symmetry character. The CEF ground state of a paramagnetic $USb_2$ Hund's metal is theoretically predicted to be a robust non-magnetic singlet, creating an exotic setting for magnetism that resembles an exciton condensate, and is previously only known from fragile and low temperature realizations. The temperature dependence of XLD measurements is found to reveal a symmetry evolution consistent with singlet-based magnetism. Neutron diffraction measurements show a relatively sharp local moment onset at the transition, consistent with the importance of fluctuations to nucleate the singlet-based magnetic transition, and suppressing thermal fluctuations in a doping series is found to quench magnetism beneath $T_N < \sim 100$K.

Taken together, these measurements are consistent with a singlet-based magnetic energy hierarchy that yields an anomalously large number of thermally accessible degrees of freedom at low temperature ($T < 100$K), and provides a foundation for explaining the otherwise mysterious coherence effects found in previous transport, heat capacity, and ARPES measurements at $T < 100$K[7,9–11]. The interchangeability of elements on both the uranium (demonstrated as $U_{1-x}Th_xSb_2$) and pnictogen site suggests $UX_2$ as a model system for exploring the crossover into both Hund's metal and singlet-ground-state magnetic regimes.

## Methods

**Experiment.** The samples of $UBi_2$ and $USb_2$ were top-posted in a nitrogen glovebox and then transferred within minutes to the ultra high vacuum (UHV) environment. The samples were cleaved in UHV and measured in-situ, with initial U O-edge spectra roughly 30 minutes after cleavage. The UV-XAS measurements were performed in the MERIXS (BL4.0.3) in the Advanced Light Source with base pressure better than $4 \times 10^{-10}$ Torr. The switch between linear horizontal polarization (LH-pol) and linear vertical polarization (LV-pol) is controlled by an elliptically polarizing undulator (EPU) and keep precisely the same beam spot before and after the switch. The incident angle of the photon beam was 30°, which gives a 75% out-of-plane E-vector spectral component under the LH-pol condition and 100% in-plane E-vector under the LV-pol condition. The XAS signal was collected by the total electron yield (TEY) method. The penetration depth of VUV and soft X-ray XAS measured with the TEY method is generally in the 2–4 nm range set by the mean free path of low energy ($E < \sim 10$ eV) secondary electrons created in the scattering process[30], making it a much more bulk sensitive technique than single-particle techniques such as angle resolved photoemission.

Air-exposed $UBi_2$ can degrade rapidly due to oxidation. No evidence of a large volume fraction of oxide or other phases was found from neutron scattering data for $USb_2$ and $UBi_2$. Possible sample oxidation was surveyed by measuring oxygen $L_1$-edge XAS via TEY for both $USb_2$ and $UBi_2$ during the uranium O-edge XAS experiments. An oxygen $L_1$-edge signal was visible at the cleaved surface of both samples, and found to have similar intensity for both $USb_2$ and $UBi_2$ samples (Supplementary Note 6).

The O-edge XAS curves observed under LH-pol and LV-pol polarization are normalized by assigning constant intensity to the integrated area of the R1 region. Spectral intensity was integrated between featureless start (95 eV) and end points (102 eV) for both $UBi_2$ and $USb_2$. The linear dichroism of the XAS in the main text is defined as:

$$I_{LD} = (I_{LH} - I_{LV})/I_{LH(max)} \qquad (1)$$

where $I_{LH(max)}$ is the XAS intensity maximum under LH-pol condition within R1 region. The monotonic temperature linear dichroism of $USb_2$ in the main text is a solid result under different data normalization process but linear dichroic rate can be influenced by some factors, for example the irreducible background in $I_{LH(max)}$. In the simulation, tuning the broadening factor is also easy to change simulated linear dichroic rate which make seriously quantitative comparison of the linear dichroism between experiment and the simulation meaningless.

Neutron diffraction measurements were performed on single crystals at the BT-7 thermal triple axis spectrometer at the NIST Center for Neutron Research[31] using a 14.7 meV energy and collimation: open - 25′ - sample - 25′ - 120′. For $USb_2$, the magnetic intensity at the (1, 0, 0.5) peak was compared to the nuclear intensity at the (1, 0, 1) peak, while the temperature dependence of the (1 1, 0.5) peak was used to calculate an order parameter. For $UBi_2$, the temperature-dependent magnetic intensity at the (1, 1, 1) peak was compared to nuclear intensity at (1, 1, 1) peak at 200K, above the Néel temperature. In both cases, an $f^2$ magnetic form factor was assumed[32].

**Atomic multiplet + mean field model (AM + MF).** Atomic multiplet calculations were performed as in ref. [16], describing $5d^{10}5f^n \rightarrow 5d^95f^{n+1}$ X-ray absorption in the dipole approximation. Hartree-Fock parameters were obtained from the Cowan code[33], and full diagonalization of the multiplet Hamiltonian was performed using LAPACK drivers[34]. Hartree-Fock parameters for $5f$ multipole interactions renormalized by a factor of $\beta = 0.7$ for $UBi_2$, and a more significant renormalization of $\beta = 0.55$ was found to improve correspondence for $USb_2$. This difference matches the expected trend across a transition between $5f^2$ and $5f^1$ local multiplet states. Core-valence multipole interactions renormalized by $\beta_C = 0.55$, consistent with other shallow core hole actinide studies[35]. The $5f$ spin orbit is not renormalized in $USb_2$ but renormalized by a factor of 1.15 in $UBi_2$ due to the much larger spin orbit coupling on bismuth. A detailed comparison of simulation results generated from two sets of Hartree-Fock parameters is included in Supplementary Note 7.

Total electron yield is dominated by secondary electrons following Auger decay of the primary scattering site. We have assigned core hole lifetime parameters to describe this decay, and adopted the common approximation that the number of secondary electrons escaping from the material following each core hole decay event is independent of the incident photon energy. For the $5f^1$ simulation, the core hole inverse lifetime is $\Gamma = 1.4$ eV at $h\nu < 100$ eV, $\Gamma = 1.8$ eV at 100 eV $< h\nu < 108.5$ eV, and 6.5 eV at $h\nu > 108.5$ eV. For $5f^2$ and $5f^3$ simulations, feature widths were obtained from a core hole inverse lifetime set to $\Gamma = 1.3$ eV ($h\nu < 99$ eV), $\Gamma = 1.5$ eV (99 eV $< h\nu < 103.5$ eV), and 6.5 eV ($h\nu > 103.5$ eV). In the $5f^3$ simulation, assigning the 103 eV XAS feature to R1 (longer lifetime) as in the Fig. 2 makes it more prominent than if it is assigned to R2 (shorter lifetime). It is also worth noting that scenarios intermediate to $5f^2$ and $5f^3$ do not necessarily closely resemble the $5f^3$ endpoint, and spectral weight in the 103 eV $5f^3$ XAS peak may depend significantly on local hybridization. However, in real materials, $5f^3$ character is associated with a downward shift in the R1 resonance onset energy that is opposite to what is observed in our data[36].

The mean field model was implemented by considering the $USb_2$ uranium sublattice with Ising exchange coupling between nearest neighbors:

$$H = \sum_i A_{,i} + \sum_{\langle i,k \rangle} J_{i,k} S_{z,i} S_{z,k} \qquad (2)$$

where $A_{,i}$ is the $5f^2$ single-atom multiplet Hamiltonian, $J_{i,k}$ is an exchange coupling parameter with distinct values for in-plane versus out-of-plane nearest neighbors, and $S_{z,i}$ is the z-moment spin operator acting on site $i$. Mean field theory allows us to replace one of the spin interaction terms ($S_{z,k}$) with a temperature-dependent expectation value, and describe the properties of the system in terms of a thermally weighted single-atom multiplet state ensemble. The specific values of individual $J_{i,k}$ terms are unimportant in this approximation, however their signs must match the antiferromagnetic structure in Fig. 1, and the sum of the absolute value of near-neighbor terms must equal $J_{eff} = \sum_{<k>} |J_{n,k}| = 43$ meV to yield a magnetic transition at $T_N = 203$ K. When considering the doped case of $U_{1-x}Th_xSb_2$, the expectation value $< S_{z,k} >$ is effectively reduced by weighting in the appropriate density of 0-moment $5f^0$ Th sites.

The CEF energy hierarchy has not been fine tuned. Perturbation strengths are scaled to set the lowest energy excitation to 10 meV, a round number that roughly matches the lowest $k_B T_N$ value at which a magnetic transition is observed in $U_{1-x}Th_xSb_2$. This assignment gives a total energy scale for crystal field physics that is approximately comparable to room temperature ($\Delta CEF \sim k_B T_N$), as expected for this class of materials, and the associated orbital energies were found to correspond reasonably (within $< \sim 30\%$) with coarse estimates from density functional theory. The crystal field parameters are listed in the first column of Table 1.

The low temperature ordered moment of $M = 1.90$ $\mu_B$ seen by neutron scattering is matched by downward-renormalizing the moment calculated in the mean field model to 62% (see Fig. 4d shading). Within density functional theory (DFT) models, the consideration of itinerant electronic states provides a mechanism to explain most of this discrepancy. In DFT simulations, the spin component of the magnetic moment is enhanced to $M_S \sim 2$ $\mu_B$[14,15], larger than the maximal value of $M_S \sim 1.4$ $\mu_B$ that we find in the $5f^2$ ($J = 4$) atomic multiplet picture. This larger DFT spin moment is directly opposed to the orbital magnetic

moment, resulting in a smaller overall ordered moment. The ordered moment in the multiplet simulation could alternatively be reduced by strengthening the crystal field, but this is challenging to physically motivate, and has the opposite effect of reducing the spin moment to $M_S < 1\,\mu_B$.

**Density functional theory + dynamical mean field theory (DFT + DMFT).** The combination of density functional theory (DFT) and dynamical mean-field theory (DMFT)[37], as implemented in the full-potential linearized augmented plane-wave method[38,39], was used to describe the competition between the localized and itinerant nature of 5f-electron systems. The correlated uranium 5f electrons were treated dynamically by the DMFT local self-energy, while all other delocalized spd electrons were treated on the DFT level. The vertex corrected one-crossing approximation[38] was adopted as the impurity solver, in which full atomic inter-action matrix was taken into account. The Coulomb interaction $U = 4.0$ eV and the Hund's coupling $J = 0.57$ eV were used for the DFT + DMFT calculations.

**Code availability.** Though the source code used for these multiplet calculations is not publicly available, there are excellent options with equivalent capabilities such as CTM4XAS (http://www.anorg.chem.uu.nl/CTM4XAS/) and Quanty (http://www.quanty.org).

## Data availability

All relevant data of this study are available from the corresponding author upon reasonable request.

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

## Acknowledgements

We are grateful for discussions with S. Roy and L. Klein. This research used resources of the Advanced Light Source, which is a DOE Office of Science User Facility under contract no. DE-AC02-05CH11231. Work at NYU was supported by the MRSEC Program of the National Science Foundation under Award Number DMR-1420073. P.A. acknowledges funding from the U.S. National Science Foundation CAREER under award No. NSF-DMR 1654482. The identification of any commercial product or trade name does not imply endorsement or recommendation by the National Institute of Standards and Technology. G.K. and C.-J.K. are supported by DOE BES under grant no. DE-FG02-99ER45761. G.K. carried out this work during his sabbatical leave at the NYU Center for Quantum Phenomena, and gratefully acknowledges NYU and the Simons foundation for sabbatical support. Y.W. was supported by the US Department of energy, Office of Science, Basic Energy Sciences as a part of the Computational Materials Science Program through the Center for Computational Design of Functional Strongly Correlated Materials and Theoretical Spectroscopy.

## Author contributions

L.M., R.B., Y.X., E.K., and H.H. carried out the XAS experiments with support from J.D. D., Y.-D.C., and J.R.J.; neutron measurements were performed by S.R. and N.P.B. with support from Y.Z., Z.X., and J.W.L.; STM measurements were performed by I.G., with guidance from P.A.; high quality samples were synthesized by S.R. and S.R.S. with guidance from N.P.B.; multiplet simulations were performed by L.M. with guidance from L.A.W., and DFT + DMFT simulations were performed by C.-J.K. with assistance from Y.W. and guidance from G.K.; L.M., R.B., Y.X., P.A., N.P.B., and L.A.W participated in the analysis, figure planning, and draft preparation; L.A.W. was responsible for the conception and the overall direction, planning, and integration among different research units.

## Additional information

**Competing interests:** The authors declare no competing interests.

