## [Peer Review File · Nature Communications]

Reviewers' comments:

Reviewer #1 (Remarks to the Author):

Prof. Wray and co-authors have presented an exciting use of the uranium O-edge XAS to probe different valence symmetries. The scientific work is well done, and I believe this work will be suitable for publication in Nature Communications.

Authors present a very detail analysis of the electronic structure of USb₂ and UBi₂ systems, based on the cutting edge experiments at the large scale facilities – ALS and NIST. Experimental data supported by variety of theoretical calculations which match very well the experimental observations. I very much appreciate the idea to show a negative second derivative (SDI) results by comparing the experimental XAS data and multiplet calculations – It looks very convincing! State of the art- AM+MF and DFT+DMFT calculations of the temperature depended experimental data look very impressive. And definitely the further discussion about different physical impact of then nominal 5f₁ and 5f₂ configuration is excellently and clearly written. This could present an exciting opportunity to explore single ground state magnetic regime from Hund's rules correlations in actinide strongly correlated systems.

However I have few comments:

- R₂ is not clearly indicated in Fig.2
- XAS spectrum of the UBi₂ is not fully plotted (only up to 112 eV) – it might be important to show entire spectrum
- I didn't really understand how authors found out that 90% of 5f₁ multiplet state is present in UBi₂ system? What about other 10%? How accurate those values are?
- It will be great to note which program has been used for the multiplet calculations
- Similar to previous comment – DFT+DMFT shows 25% of admixture of 5f₁ and 5f₃ configurations. Where is it coming from?
- Results about Oxygen L₁ edge are not shown. Could you please include them?
- The statement about 15% of intensity difference for cleaved sample is not clear to me. Does it mean that there is still Oxygen present at the surface of the non-cleaved sample?
- Going through the Methods – I don't really understand the reason to use different Hartree-Fock parameters for 5f₁ and 5f₂ configurations. Perhaps it will be interesting to show in SI materials how 5f₁ and 5f₂ multiplets look if calculations were done with identical parameters.
- I also went through the literature search about multiplet calculations, and it seems to be a long history there. Kotani and Ogasawara (*Physica B*, 186-188, 16, 1993) showed 5f₂ and 5f₃ calculations and claim that effects of hybridization and configuration interactions are very crucial for the 5f₃ conf. Did you take into account?
- Additionally the calculations of 5f₃ configurations, reported by Kotani looks different to the one reported here. Could you please comment on it?
- There are few more papers, who reported already XAS 5d multiplet calculations previously. It is worth to compare your results this results reported previously. For example - A chapter in book "Actinide Nanoparticle Research" 2011 by S. Butorin, where he shows plenty of calculations and details about it. Plus Butorin et al. *Anal.Chem.* 85 ,11196 -11200, 2013.

Reviewer #2 (Remarks to the Author):

The manuscript reports the results of high quality XAS measurements of the O edge of U in USb₂ and UBi₂. UBi₂ is identified as being mostly f₁ but the data shows that USb₂ is found to be either itinerant or mixed valent with a large f₂ component to the state.

The results are interesting and deserves publication.

The paper is weakened by a speculative interpretation of the magnetic properties of USb₂. The model is atomic model with mean field magnetic interactions that describes a singlet ground state which undergoes a transition to a magnetic state by mixing with excited magnetic states. Such scenarios have been discussed before in other contexts and is thrown in here without much justification.

Despite the weak interpretation, I recommend publication.

Reviewer #3 (Remarks to the Author):

The manuscript „High temperature singlet-based magnetism from Hund’s rule correlations“ by Lin Miao and co-authors investigates electronic properties and magnetism in two uranium based materials USb₂ and UBi₂ by performing a comparative analysis.

I would like to mention that there are only a few working groups worldwide dealing with electronic properties of uranium or, more generally, actinide systems. So, the community is very small, but belongs to the rather large community that deals with strong electronic correlations. Uranium systems, where the 5f bandwidth is only slightly smaller than the on-site Coulomb correlation energy, form the bridge between Kondo physics and mixed valence in localized 4f systems and correlated late transition metals with predominantly itinerant d-bands. In uranium systems the localized and itinerant character of the 5f electrons have to be discussed at eye level, which makes the treatment difficult, but the systems even more interesting. Thus, a publication in Nature Comm. of the presented results is definitely worth considering.

The authors use XAS at the U-O edge in total yield mode to derive an effective local 5f occupation for the compounds USb₂ and UBi₂ from the observed XAS multiplet structures. As a result they find 5f₁ for UBi₂ and 5f₂ for USb₂, whereby the latter is largely in agreement with the result of a DFT-DMFT calculation. In the crystal electric field this results in a non-magnetic singlet ground state for USb₂, which is rather seldom in nature, but allows to explain the unusual magnetic properties as excitation phenomena of this non-magnetic ground state, which among others are reflected in the temperature dependence of the XAS signal and neutron-scattering data. This is a fascinating magnetic system, especially because excitation phenomena, which occur in a few rare earth compounds only at extremely low temperatures, can be observed here under moderate conditions. The given explanations sound quite interesting and reasonable and the results are without question exciting what could justify publication in Nature Comm.

To determine the effective 5f_n occupation, the authors compare the measured near edge structure with the results of a simulation. Unfortunately, the authors give relatively few details on the latter: At the 5d excitation threshold an excitation 5d₁₀5f_n→5d₉5f_{n+1} takes place which decays predominantly into a 5d₁₀5f_{n-1} final state, which essentially contributes to the total electron yield measurement. Here it would be important to indicate what was calculated in detail. The values of n given in the figure correspond to the ground state, which is not directly mapped in the experiment.

For the non-experts in actinide magnetism among the readers it would also be helpful to show an energy scheme for the expected CEF splitting of the 5f₂ state and to illustrate the excitations discussed in the text not only by words and a table, but on the basis of this scheme.

There are also a few minor points. The resonance R₂ is discussed together with the R₁ feature, however, R₂ is not shown in figure 2a, and should be included. In the abstract, from the sentence “The evolution of symmetries across...” it is unclear what kind of symmetries the authors are discussing here. Also, the reference to the Supp. Notes and figures are not in the correct order. For example, the first reference on the 5th line addresses the Supp. Note 5.

Finally, I would recommend publication of this manuscript after the above-mentioned points have been included in suitable form.

We are very grateful for the review of our manuscript, and happy that all three referees have found the work to be of interest. All referee remarks have been addressed in the revised submission, and we have appended source data for each figure to comply with the new policy of the journal. The full referee report is reproduced below, with our responses in red.

Reviewer #1 (Remarks to the Author):

Prof. Wray and co-authors have presented an exciting use of the uranium O-edge XAS to probe different valence symmetries. The scientific work is well done, and I believe this work will be suitable for publication in Nature Communications.

Authors present a very detail analysis of the electronic structure of USb₂ and UBi₂ systems, based on the cutting edge experiments at the large scale facilities – ALS and NIST. Experimental data supported by variety of theoretical calculations which match very well the experimental observations. I very much appreciate the idea to show a negative second derivative (SDI) results by comparing the experimental XAS data and multiplet calculations – It looks very convincing! State of the art- AM+MF and DFT+DMFT calculations of the temperature depended experimental data look very impressive. And definitely the further discussion about different physical impact of then nominal 5f₁ and 5f₂ configuration is excellently and clearly written. This could present an exciting opportunity to explore single ground state magnetic regime from Hund's rules correlations in actinide strongly correlated systems.

However I have few comments:

- R2 is not clearly indicated in Fig.2

Author reply: Thank you for pointing out this. We have added a light green shaded area to indicate R2 in Fig. 2(a) of the revised manuscript.

- XAS spectrum of the UBi₂ is not fully plotted (only up to 112 eV) – it might be important to show entire spectrum

Author reply: We have now extended the XAS spectrum of UBi₂ to 117eV to cover the main feature of R2 in the revised manuscript. Unfortunately, we do not have data from higher energy than this. We performed the same XAS measurement a number of times on different samples and beam spots to ensure reproducibility, and the measurement was typically cut off at 112eV for the sake of speed.

- I didn't really understand how authors found out that 90% of 5f₁ multiplet

state is present in UBi₂ system? What about other 10%? How accurate those values are?

Author reply: We are grateful to the referee for raising this question. The text has now been revised to remove this number, and merely say that the “*lack of prominent 5f² multiplet features suggests that the 5f¹ multiplet state is quite pure*”.

The “~>90%” estimate was made based on an evaluation of the theoretical spectra and the experimental noise threshold. No clear 5f² XAS feature (98.2eV, Peak B) was found in UBi₂ spectrum under condition where the noise to signal is less than 10%. However in retrospect we believe that there is no very good quantitative approach to create a meaningful number here, in the absence of a set of curves showing the trend from one extreme to the other.

- It will be great to note which program has been used for the multiplet calculations

Author reply: We have added a citation to the Cowan code, which was used to obtain Hartree-Fock parameters, and the LAPACK drivers used for diagonalization.

The multiplet Hamiltonian and photon matrix elements were generated via a code we maintain in-house, which has been used in at least 18 published papers since 2012. We don't advertise it much, because there are very good freeware options such as CTM4XAS and Quanta that are exactly identical in this respect.

- Similar to previous comment – DFT+DMFT shows 25% of admixture of 5f¹ and 5f³ configurations. Where is it coming from?

Author reply: Metal-ligand hybridization forbids a pure 5f² configuration in the f-orbitals, even if the per-site electronic degrees of freedom come entirely from the 5f² multiplet basis. The off-valence components come from metal-ligand hybridization, with some contribution from inter-site hopping. This is discussed in Supplementary Note 1, and we have added a note that “the DFT+DMFT valence histogram restricted to a single-atom basis (as in Fig. 4a of the main text) shows that the occupancy of the U_{Sb₂} U 5f orbital is 2.17, representing a nominal 5f² valence with weak mixed-valence character due largely to the metal ligand hybridization.”

- Results about Oxygen L1 edge are not shown. Could you please include them?

Author reply: A new figure (Fig. S5) and note (Note 6) have been added in the Supplementary Information to present the oxygen L₁-edge data.

- The statement about 15% of intensity difference for cleaved sample is not clear to me. Does it mean that there is still Oxygen present at the surface of the non-cleaved sample?

Author reply: The 15% attribution has been removed from the main text, and is instead assessed at greater length in the supplement (see Fig. S5), together with fits of the oxygen L₁-edge data.

The role of oxygen on uncleaved surfaces is difficult to address, as we have not performed measurements on uncleaved samples. The neutron measurements were inconsistent with a large volume fraction of any second phase (e.g. an oxide), and the samples were kept in anaerobic environments, so we are relatively confident that there was minimal/negligible oxygen contamination prior to the XAS measurements. Cleaved surfaces of USb₂ and UBi₂ are polar, and one expects a certain amount of oxygen to show up in the form of adsorbed polar molecules that are present in the vacuum (H₂O, CO, CO₂).

- Going through the Methods – I don't really understand the reason to use different Hartree-Fock parameters for 5f₁ and 5f₂ configurations. Perhaps it will be interesting to show in SI materials how 5f₁ and 5f₂ multiplets look if calculations were done with identical parameters.

Author reply: The difference is minimal, but adjusting the Hartree-Fock parameters improves correspondence with feature energies. A new figure with both scenarios (Fig. S6) and note (Note 7) have been added in the Supplementary Information to show that the analysis does not depend on this detail.

- I also went through the literature search about multiplet calculations, and it seems to be a long history there. Kotani and Ogasawara (Physica B, 186-188, 16, 1993) showed 5f₂ and 5f₃ calculations and claim that effects of hybridization and configuration interactions are very crucial for the 5f₃ conf. Did you take into account?

Author reply: We have added a note citing the Kotani and Ogasawara 1993 paper to explain that “scenarios intermediate to 5f² and 5f³ do not necessarily closely resemble the 5f³ endpoint, and spectral weight in the 103 eV 5f³ XAS peak may depend significantly on local hybridization. However, in real materials, 5f³ character is associated with a downward shift in the R1 resonance onset energy that is opposite to what is observed in our data

[Kotani and Ogasawara, Physica B, 186-188, 16, 1993].”

We have simulated some hybridized scenarios, but hybridization was ultimately not included in the paper for the sake of simplicity. It was our feeling that the gains from adding in hybridization are counterbalanced by greater complexity, and the fact that one is potentially using an incorrect approach to compensate for missing factors such as backbonding and intersite itinerancy/screening.

- Additionally the calculations of $5f^3$ configurations, reported by Kotani looks different to the one reported here. Could you please comment on it?

Author reply: The only qualitatively important difference with our simulation is in the choice of core hole lifetime broadening parameters. We have reproduced Kotani's calculation in Fig. R1(left) below as a basis for comparison. The key issue is whether the 103 eV XAS feature is lumped with the R1 resonance (longer lifetime) or with R2 (shorter lifetime). The importance of lifetime broadening parameters for the $5f^3$ spectrum is now noted in the text, and a description of this particular issue has been added in the Methods section.

Figure R1: (left) The $5f^3$ XAS simulation from Ref. [Kotani and Ogasawara, Physica B, 186-188, 16, 1993] is very similar to (right) our simulated $5f^3$ spectrum with a broadening threshold shifted to beneath the 103 eV feature. This strong similarity occurs in spite of real (but minor) differences between the input parameters used for the earlier calculation. (The two calculations use different crystal fields, Slater-Condon renormalization, and final state broadening)

- There are few more papers, who reported already XAS $5d$ multiplet calculations previously. It is worth to compare your results this results reported previously. For example - A chapter in book "Actinide Nanoparticle Research" 2011 by S. Butorin, where he shows plenty of calculations and details about it.

Plus Butorin et al. Anal.Chem. 85 ,11196 -11200, 2013.

Author reply: We are grateful for the suggestion. We looked into [Butorin et al. Anal. Chem. 85 ,11196 -11200, (2013)] and tracked down the origin of the simulation in [J. Electron Spectrosc. Relat. Phenom. 110 –111 (2000) 213 – 233], a paper we have previously referenced in discussing uranium $5f^2$ resonance (see [PRL 114, 236401]). We have added a citation to this JESRP paper to note that it identifies the same prominent leading edge feature at the onset of a $\sim 3\text{eV}$ wide R1 resonance plateau as a characteristic indicator of $5f^2$ valence.

These papers by Butorin et al explore only the R1 edge, and use essentially the same multiplet modeling parameters as Kotani's earlier work. The UO_2 spectrum is essentially the same as what we've shown, with four main sub-features in the R1 region.

Reviewer #2 (Remarks to the Author):

The manuscript reports the results of high quality XAS measurements of the O edge of U in USb₂ and UBi₂. UBi₂ is identified as being mostly f₁ but the data shows that USb₂ is found to be either itinerant or mixed valent with a large f₂ component to the state.

The results are interesting and deserves publication.

The paper is weakened by a speculative interpretation of the magnetic properties of USb₂. The model is atomic model with mean field magnetic interactions that describes a singlet ground state which undergoes a transition to a magnetic state by mixing with excited magnetic states. Such scenarios have been discussed before in other contexts and is thrown in here without much justification.

Despite the weak interpretation, I recommend publication.

Author reply: We are grateful for the recommendation, and that the referee found the work to be of interest. We agree with the sentiment that this kind of atomic multiplet model with mean field interactions is quite limited. Nonetheless, it fills a useful role as a vehicle to roughly evaluate matrix elements in the presence of a magnetic perturbation, and to frame certain considerations for discussion. We attempted to mitigate the model's limitations by pointing out areas in which the physics can be expected to deviate from the model, such as in the magnetic critical exponent and in low temperature superexchange/Kondo-like phenomenology. Some additional discussion of earlier multiplet numerics for uranium has been added to the revised text (see response to Ref. 1).

Reviewer #3 (Remarks to the Author):

The manuscript “High temperature singlet-based magnetism from Hund’s rule correlations“ by Lin Miao and co-authors investigates electronic properties and magnetism in two uranium based materials USb₂ and UBi₂ by performing a comparative analysis.

I would like to mention that there are only a few working groups worldwide dealing with electronic properties of uranium or, more generally, actinide systems. So, the community is very small, but belongs to the rather large community that deals with strong electronic correlations. Uranium systems, where the 5f bandwidth is only slightly smaller than the on-site Coulomb correlation energy, form the bridge between Kondo physics and mixed valence in localized 4f systems and correlated late transition metals with predominantly itinerant d-bands. In uranium systems the localized and itinerant character of the 5f electrons have to be discussed at eye level, which makes the treatment difficult, but the systems even more interesting. Thus, a publication in Nature Comm. of the presented results is definitely worth considering.

The authors use XAS at the U-O edge in total yield mode to derive an effective local 5f occupation for the compounds USb₂ and UBi₂ from the observed XAS multiplet structures. As a result they find 5f₁ for UBi₂ and 5f₂ for USb₂, whereby the latter is largely in agreement with the result of a DFT-DMFT calculation. In the crystal electric field this results in a non-magnetic singlet ground state for USb₂, which is rather seldom in nature, but allows to explain the unusual magnetic properties as excitation phenomena of this non-magnetic ground state, which among others are reflected in the temperature dependence of the XAS signal and neutron-scattering data. This is a fascinating magnetic system, especially because excitation phenomena, which occur in a few rare earth compounds only at extremely low temperatures, can be observed here under moderate conditions. The given explanations sound quite interesting and reasonable and the results are without question exciting what could justify publication in Nature Comm.

To determine the effective 5f_n occupation, the authors compare the measured near edge structure with the results of a simulation. Unfortunately, the authors give relatively few details on the latter: At the 5d excitation threshold an excitation $5d105f_n \rightarrow 5d95f_{n+1}$ takes place which decays predominantly into a $5d105f_{n-1}$ final state, which essentially contributes to the total electron yield measurement. Here it would be important to indicate what was calculated in detail. The values of n given in the figure correspond to the ground state, which is not directly mapped in the experiment.

Author reply: In the revised manuscript, we have added text within the Methods section explicitly describing the states and transitions that were calculated, and explaining that the total electron yield is dominated by

secondary electrons following Auger decay. We have further noted that our calculation “adopts the common approximation that the number of secondary electrons escaping from the material following each core hole decay event is independent of the incident photon energy.”

For the non-experts in actinide magnetism among the readers it would also be helpful to show an energy scheme for the expected CEF splitting of the $5f^2$ state and to illustrate the excitations discussed in the text not only by words and a table, but on the basis of this scheme.

Author reply: We have added a figure to the supplement (Fig. S4 and Supplementary Note 5) showing the $5f^2$ CEF energy levels, and providing some additional discussion of how they relate to magnetism.

There are also a few minor points. The resonance R2 is discussed together with the R1 feature, however, R2 is not shown in figure 2a, and should be included. In the abstract, from the sentence “The evolution of symmetries across...” it is unclear what kind of symmetries the authors are discussing here. Also, the reference to the Supp. Notes and figures are not in the correct order. For example, the first reference on the 5th line addresses the Supp. Note 5.

Author response: Thank you for pointing out these issues. They have been addressed as follows:

- 1) We have added light green shaded area indicating the R2 resonance in Fig. 2(a).
- 2) The abstract now more clearly identifies the relevant symmetries as follows, “The evolution of **crystal field symmetries and magnetic ordered moment** across...”
- 3) The Supplementary Notes and Figures have been reordered to match the order of references in the main text.

Finally, I would recommend publication of this manuscript after the above-mentioned points have been included in suitable form.

REVIEWERS' COMMENTS:

Reviewer #1 (Remarks to the Author):

Thank you very much for addressing all comments. I recommend the present manuscript for the publication.

Reviewer #3 (Remarks to the Author):

I have reviewed the revised version of the paper «High temperature singlet-based magnetism from Hund's rule correlations», by Lin Miao and collaborators. I am satisfied with the answers the authors have given as well as with the performed revision of the manuscript and supplementary info file. I therefore recommend the paper for publication in Nature Communications.